# A real-time assay for cell-penetrating peptide-mediated delivery of molecular cargos

**Schuyler B. Gentry**[1¤a], **Scott J. Nowak**[1], **Xuelei Ni**[2], **Stephanie A. Hill**[1], **Lydia R. Wade**[1], **William R. Clark**[1], **Aidan P. Keelaghan**[1¤b], **Daniel P. Morris**[1], **Jonathan L. McMurry**[1]*

**1** Department of Molecular & Cellular Biology, Kennesaw State University, Kennesaw, Georgia, United States of America, **2** School of Data Science & Analytics, Kennesaw State University, Kennesaw, Georgia, United States of America

¤a Current address: $Q^2$ Solutions, Atlanta, Georgia, United States of America
¤b Current address: Emory University School of Medicine, Atlanta, Georgia, United States of America
* jmcmurr1@kennesaw.edu

**Data Availability Statement:** All relevant data are within the manuscript and its Supporting Information files.

**Funding:** This work was supported by National Institute on Biomedical Imaging and Engineering

## Abstract

Cell-penetrating peptides (CPPs) are capable of transporting molecules to which they are tethered across cellular membranes. Unsurprisingly, CPPs have attracted attention for their potential drug delivery applications, but several technical hurdles remain to be overcome. Chief among them is the so-called 'endosomal escape problem,' i.e. the propensity of CPP-cargo molecules to be endocytosed but remain entrapped in endosomes rather than reaching the cytosol. Previously, a CPP fused to calmodulin that bound calmodulin binding site-containing cargos was shown to efficiently deliver cargos to the cytoplasm, effectively overcoming the endosomal escape problem. The CPP-adaptor, "TAT-CaM," evinces delivery at nM concentrations and more rapidly than we had previously been able to measure. To better understand the kinetics and mechanism of CPP-adaptor-mediated cargo delivery, a real-time cell penetrating assay was developed in which a flow chamber containing cultured cells was installed on the stage of a confocal microscope to allow for observation *ab initio*. Also examined in this study was an improved CPP-adaptor that utilizes naked mole rat (*Heterocephalus glaber*) calmodulin in place of human and results in superior internalization, likely due to its lesser net negative charge. Adaptor-cargo complexes were delivered into the flow chamber and fluorescence intensity in the midpoint of baby hamster kidney cells was measured as a function of time. Delivery of 400 nM cargo was observed within seven minutes and fluorescence continued to increase linearly as a function of time. Cargo-only control experiments showed that the minimal uptake which occurred independently of the CPP-adaptor resulted in punctate localization consistent with endosomal entrapment. A distance analysis was performed for cell-penetration experiments in which CPP-adaptor-delivered cargo showing wider dispersions throughout cells as compared to an analogous covalently-bound CPP-cargo. Small molecule endocytosis inhibitors did not have significant effects upon delivery. The real-time assay is an improvement upon static endpoint assays and should be informative in a broad array of applications.

(https://www.nibib.nih.gov/) grant EB028609 (to JLM). SJN is supported by AHA grant 18AIREA33960109 (www.heart.org). SBG was supported by a graduate research assistantship from the KSU Graduate College. The funders had no role in study design, data collection and analysis, decision to publish, or preparation of the manuscript.

## Introduction

Many promising therapeutic leads fail because they cannot reach their targets, of which frequent causes are failure to cross cell membranes and traffic to appropriate subcellular destinations. Cell-penetrating peptides (CPPs) have long held great promise for transmembrane delivery of therapeutics precisely because they can overcome these problems [1–4]. They catalyze transmembrane passage of a wide variety of biomolecules with vastly different physicochemical characteristics from large, oligomeric proteins to nucleic acids and small molecules. Generally, if a molecule (the 'cargo') can be fused to a CPP, it can be delivered into a cell. CPP-containing molecules are rapidly taken up by a wide array of eukaryotic cell types. Though dozens of clinical trials are underway [3, 5], development of CPP-based therapeutics has been hampered several vexing problems including toxicity at high doses currently required, a lack of ability to target specific cells and covalent or otherwise irreversible CPP-cargo linkages that result in entrapment in endosomes, the so-called 'endosomal escape problem' [6]. No CPP-based therapeutics have yet been approved and several clinical trials have been terminated, but hope springs eternal and recent advances have been made to address the technical hurdles yet to be overcome [7, 8].

Most CPP delivery schemes utilize irreversible linkages to cargo, usually covalent, otherwise nonspecific or hydrophobic [9]. We previously reported the development and application of CPP-tagged adaptor proteins with reversible noncovalent, $Ca^{2+}$-dependent couplings that enable cargos to be transported into cells and then released from the CPP [10, 11]. Our lead adaptor, TAT-CaM, consists of the cell-penetrating moiety of HIV transactivator of transcription (TAT) N-terminally fused to human calmodulin (CaM). Cargo proteins are recombinant fusions containing a 17-residue calmodulin binding site (CBS). TAT-CaM/CBS-cargo complexes spontaneously bind with nM affinity in the presence of $Ca^{2+}$, but negligibly in its absence. Changes in $[Ca^{2+}]$ and pH after internalization [8] result in release of cargo. We have shown that numerous cargos reach the cytoplasm or other subcellular destination in multiple cell lines once released from the CPP-adaptor even though the CPP itself remains trapped in endosomes [9, 10], i.e. we have solved the endosomal escape problem. Our technology gets cargo into cells and out of endosomes into the cytoplasm even though the CPP itself, like others, remains trapped. Increased escape efficiency may allow for lower dosing. Other CPP-cargos must be present in μM or tens of μM concentrations to observe cytoplasmic delivery whereas TAT-CaM achieves observable delivery at nM concentrations.

Optimization of the CPP-EF-hand architecture of TAT-CaM and other adaptors has also been a priority. It was previously reported that several other EF-hand proteins and several other CPP moieties evinced no discernible difference in delivery of a model cargo [11], suggesting that calmodulin was at least as effective as other EF hand proteins. In the present study, we examined calmodulins from other species for qualities that would increase efficacy in delivery, finding that naked mole rat (*Heterocephalus glaber*) calmodulin ("NMR-CaM") was more effective at inducing internalization than human calmodulin. NMR-CaM varies from human in several respects, most dramatically in molecular weight (~30 kDa vs ~17 kDa) and net charge (-15 vs. -24) but binds a human calmodulin binding moiety with similar affinity and kinetics. It has an amino-terminal domain of unknown function lacking discernible homology to known sequences from residues 1–128 but the rest is invariant from human calmodulin save the analogous position of the initiating methionine, 129, is a leucine in NMR-CaM.

A number of worthy efforts have been made to examine the kinetics of CPP-mediated cargo delivery [12–17]. These studies were examinations of different CPPs in various cell lines under different conditions, and they used various methods of detection that relied on analyzing whole cells, including radiometric and spectrophotometric assays, flow cytometry and

conventional fluorescence microscopy. While a few informative studies have been done using exotic techniques that measured actual cytosolic delivery, e.g. $^{19}$F NMR [18], it has long been desirous to have a relatively simple assay that analyzes delivery to cell interiors in living cells using readily labeled cargos. In this study, we developed a real-time, live cell assay using commercially available channel slides that allows cells to be cultured in a flow chamber so that direct imaging of cell interiors during cargo delivery can be done using confocal microscopy. Using a peristaltic pumping system, CPP-cargo complexes were applied to cells from a reservoir to effect subcellular observation from time zero under buffered and temperature-controlled conditions.

We were interested not only in the kinetics of CPP-mediated delivery of cargo, but also in better understanding how our CPP-adaptors' $Ca^{2+}$-mediated release works inside the cell, which required improving our static endpoint assay to be able to observe early time periods during the course of delivery. Under conditions used in this study, TAT-NMR-CaM delivered a model cargo, maltose binding protein, to the cytoplasm of baby hamster kidney (BHK) cells with observable fluorescence in seven minutes. In contrast to some earlier work, delivery was linear over the course of the assay, perhaps due to the much lower dosing our CPP-adaptors enable. Distance analysis of delivery with comparison to an analogous covalently conjugated TAT-cargo protein showed less punctate cellular distribution, consistent with enhanced endosomal escape. Addition of endocytosis inhibitors was largely inconclusive, but treatment with methyl-β-cyclodextrin, which inhibits caveolin-mediated endocytosis, appeared to reduce delivery, though only at substantially high concentrations that precluded strong conclusions. Future studies will examine dosing, temperature and other parameters as well as other cargos, CPPs and endosomal escape enhancers to better understand the mechanism and kinetics of delivery.

Here we report the first real-time assay using direct observation of cargo delivery to the cytoplasm of living cells by confocal microscopy. It is an improvement upon earlier kinetic assays in that unlike conventional fluorescence microscopy, allows imaging at specific depths in the cell, thus overcoming problems posed by potentially confounding artifacts such as surface adherence.

## Materials & methods

### Plasmids, strains and cell lines

Plasmids used were previously described [10, 11] or constructed as described in (S1 Fig). Briefly, pJM161 consisted of an *E. coli*-optimized synthetic gene (GeneScript, Piscataway, NJ, USA) encoding TAT-naked mole rat calmodulin (TAT-NMR-CaM) cloned into NdeI and BamHI sites in pET19b (EMD Millipore, USA) with an in-frame stop codon prior to the BamHI site. The encoded TAT-NMR-CaM protein consists of the TAT peptide sequence (YGRKKRRQRRR) N-terminally fused to *Heterocephalus glaber* (naked mole rat) calmodulin (GenBank: EHB02604.1) [19]. A vector-encoded 10xHis tag is N-terminal to TAT. Plasmids pJM140 and pJM168, encoding CBS-maltose binding protein (CBS-MBP) and TAT-maltose binding protein (TAT-MBP), respectively, were made by cloning synthetic gene fragments encoding the CBS or TAT peptide sequences into NdeI and BamHI sites in pMAL-c5x (New England Biolabs, Ipswich, MA). The encoded cargo proteins thus have either a CBS or TAT sequence C-terminal to the MBP and a 6x His tag beyond.

*E. coli* used in this study, BL21(DE3)pLysS was propagated from purchased cells from EMD Millipore (Burlington, MA, USA) or other established supplier.

Baby hamster kidney (BHK) cells were purchased from ATCC (#CCL-10) and cultured in Dulbecco's Modified Eagles' Medium with GlutaMAX Supplement (Gibco, USA) and 10% fetal bovine serum.

## Expression, purification and labeling

Proteins were expressed essentially as described [20] with minor modifications. Briefly, plasmids were freshly transformed into BL21(DE3)pLysS. Overnight cultures grown from single colonies were subcultured into 1L Luria-Bertani broth and grown with vigorous shaking at 37˚C. At $OD_{600}$ ~0.4, temperature was lowered to 30˚C and cells were induced with 0.2 mM IPTG and growth continued for four hours. The procedure was altered slightly for CBS-MBP, which was grown in Terrific Broth (TB) supplemented with 0.25% (w/v) glucose and induction was conducted at 32˚ C. Cells were harvested by centrifugation at 10,000 x g and frozen at -80˚C.

Purification was also performed essentially as described [15] via immobilized metal affinity chromatography. Briefly, cell pellets were thawed on ice, resuspended in lysis buffer (50 mM Tris pH 8, 500 mM NaCl, 10 mM imidazole, 10% glycerol and 6 mM β-mercaptoethanol). 1 mg/ml DNAse and 0.25 mg/ml lysozyme were added during resuspension. For TAT-NMR--CaM only, Halt Protease Inhibitor Cocktail (ThermoFisher) was added to 1x per manufacturer's protocol. Cells were broken via passage through a French press at 20,000 psi and subjected to centrifugation at ~27,000 x g to pellet unbroken cells and debris. Clarified lysate was passed over a cobalt affinity column using an FPLC system while monitoring $A_{280}$, washed with wash buffer (equivalent to lysis buffer with 25 mM imidazole instead of 10 mM) until baseline absorbance was attained, after which protein was eluted in elution buffer (lysis buffer with 250 mM imidazole). Protein-containing fractions were pooled, concentrated and exchanged by passage over a desalting column into 10 mM HEPES, pH 7.4, 150 mM NaCl, 10% glycerol, 1 mM $CaCl_2$ for fluorescent labeling, biotinylation or other further use. Quantitation was done using Bradford Assay with bovine serum albumin as standard.

Biotinylation was accomplished using NHS-LC-LC biotin, crosslinked per the manufacturer's protocol (ThermoFisher, USA). For fluorescent labeling, DyLight 650 was similarly crosslinked to cargo proteins or TAT-CaM and dye removal columns were used to remove unreacted dye (ThermoFisher). Stoichiometries were adjusted so that on average, less than one fluorophore per protein was incorporated.

## Biolayer interferometry

Kinetic analysis was performed on ForteBio Octet Red 96 BLI biosensor (Sartorius, Göettingen, Germany) at 30˚ C. All buffer used for sample dilutions and assays was 10 mM HEPES, pH 7.4, 150 mM NaCl, 10% glycerol, 1 mM $CaCl_2$ 0.1% Tween-20. Biotinylated TAT-NMR--CaM was loaded onto streptavidin (SA) sensors. After establishing a baseline, sensors were moved into buffer containing CBS-MBP. Association was monitored for 300s after which sensors were moved to buffer only for 300 s and then buffer containing 10 mM EDTA to monitor dissociation in the presence and absence of available $Ca^{2+}$. Association and no-EDTA dissociation were fit to a one-state simple association-then-dissociation global model to determine kinetic and affinity constants. EDTA-induced dissociation was fit to a global one-state exponential decay model to assign a fast-off, $Ca^{2+}$ free dissociation constant. GraphPad Prism was used for all analysis.

## Cell penetration assays

Endpoint assays were performed as described [10, 11] in which the CPP-adaptor/cargo complexes (1 μM each component as determined by quantitation via Bradford Assay, unless noted otherwise) were incubated with cells for 1 h prior to imaging. For endocytosis inhibition assays, cells were seeded into an 8-well plate 20 hours prior so that ~60% confluency was achieved by time of imaging. Cells were pretreated for 30 min with media containing 25 μM

Chlorpromazine hydrochloride (CPZ), 5 mM methyl-B-cytodextrin (MβCD), 50 μM EIPA), 5-(N-ethyl-N-isopropyl) amiloride (EIPA) or an equal volume of solvent in the negative control sample. Inhibitor-containing media was aspirated and replaced with media containing TAT-NMR-CaM/CBS-MBP complexes in which the CBS-MBP was labelled with DyLight 650. After a 45 min incubation, cells were washed twice with phosphate-buffered saline containing Mg2+ and Ca2+ followed by addition of imaging media (DMEM Glutamax + 10% FBS + 25 mM HEPES, pH. 7.4 and NucBlue stain). Cells were then imaged at 37˚ C without $CO_2$ on the confocal microscope.

To allow for imaging during the exposure of cells to adaptor/cargo complexes, a real-time assay was developed. BHK cells were maintained in a 37˚C, 5% $CO_2$ environment with DMEM GlutaMax media (+4.5g/L D-glucose, no sodium pyruvate) supplemented with 10% fetal bovine serum (FBS) in a T-25 culture flask. Cells were prepared 20–24 hours prior to assay by deposition of approximately $4.2 \times 10^4$ cells into an Ibidi (Gräfelfing, Germany) μ-Slide VI 0.4 channel slide, which yielded approximately 70% confluency by time of assay. Media were added to channel reservoirs to reach 75% capacity and cells were incubated overnight. Thirty minutes prior to imaging, the channel slide was aspirated and fresh media containing NucBlue (ThermoFisher) nuclear stain were delivered into the channel slide. The channel slide was transferred to an incubated stage (37˚C) on a Zeiss LSM 900 confocal microscope. CPP-adaptor/CBS-cargo complexes or analogous control proteins were pumped into tubing by peristaltic pump at 0.4 ml/min using a syringe reservoir (S2 Fig). Imaging with 40X Plan-Apochromat objective (N.A. = 0.95, M27) began as proteins flowed into channel slide. Flow of protein was halted in channel after one minute and imaging continued for 40 minutes with protein present on cells during imaging process. The cells were washed with 3 mL of DMEM GlutaMax (+4.5g/L D-glucose, -sodium pyruvate) + 10% FBS + 25mM HEPES in flow as imaging continued for 10–15 minutes. After imaging was complete, 70% ethanol was pumped through tubing to clean the system. Lasers (555 nm and 405 nm) were used at 0.2% power to capture labeled protein or nuclei. Pinhole size was set at one Airy Unit (AU) for both lasers. The mode of collection was programmed to repeat per line and intensity signatures were determined based off mean intensity values. The Definite Focus feature was used continuously to maintain a consistent focus plane for image acquisition, which was located at the midpoint of the cell to assure that observed cellular fluorescence was internal and not surface adherent.

## Statistics

To determine the extent of cargo diffusion throughout the interiors of cells, a pairwise analysis was performed to measure distances between fluorescent pixels in micrographs taken from endpoint cell penetration assays at the midpoint of cells. 50 randomly selected cells for each condition were examined. Cells were outlined using ImageJ and from within those boundaries, distance measurements for all pixels that exhibited fluorescence from every other such pixel were measured. For computational reasons, distance pairs were randomly sampled to reach a data set of 10% of the original size. Using SAS® (Cary, NC, USA), values were compiled into a distance matrix and then all distances calculated were aggregated into one column, from which milliles (0.1% percentile, 0.2% percentiles, ...100% percentiles) were found and used to represent the pair-wise distances of the exhibited fluorescence in one cell. The resultant sets consisted of 50,000 points for each condition (50 cells x 1,000 milliles each) and compared in their distribution. Significance for distribution between distances in control and experimental cells was determined by Kolmogorov-Smirnov test [21].

## Results and discussion

### Purification and characterization of TAT-NMR-CaM

Hypothesizing that size and net charge would result in differences in internalization and release of CBS-containing cargos, we examined numerous calmodulin sequences in the protein database. *Heterocephalus glaber* calmodulin offered an interesting variation in size and net charge relative to human calmodulin as well as what amounted to a natural spacer element consisting of residues 1–128 between the N-terminal TAT sequence and the conserved calmodulin sequence (Fig 1). Residues 1–128 also reduced the overall net charge by 9 relative to CaM with the addition of 16 basic residues and 7 acidic ones. Expression and purification were readily achieved and TAT-NMR-CaM bound the calmodulin binding sequence common to all cargo proteins we have examined to date.

Cargo proteins used in this study differ from our previous reports. In a search for more idealized cargo, maltose binding protein (MBP) was chosen because of its high stability, solubility and very high levels of expression from pMAL. Indeed, the two MBP constructs, CBS-MBP and TAT-MBP, were produced in mg quantities per batch via a one-step affinity purification.

TAT-NMR-CaM binding to CBS-MBP exhibited complex kinetics in BLI experiments due to anomalies in the dissociation phase in which the nm shift increased despite no free ligand in solution (Fig 2A), preventing a direct determination of $k_d$ and hence $K_D$ via fit to an association-then-dissociation model. Why the anomalous behavior was observed is unknown and could include slowly-occurring oligomerization, binding-induced conformational changes or surface-induced denaturation (a phenomenon previously observed with entirely different binding partners, [22]), but it was repeatedly observed with CBS-MBP when used as either ligand or analyte. Fig 2A shows a background-subtracted sensorgram for a complete experiment in which biotinylated CBS-MBP was tethered to sensors and TAT-NMR-CaM was analyte. The partners bound rapidly (0-300s), remained tightly bound during the dissociation phase (300-1200s) and then rapidly dissociated when sensor-bound complexes were exposed to 10 mM EDTA (1200-1500s). TAT-NMR-CaM exposed to sensors without tethered CBS-MBP yielded an amplitude that was a small fraction of total binding (S3 Fig) and thus the

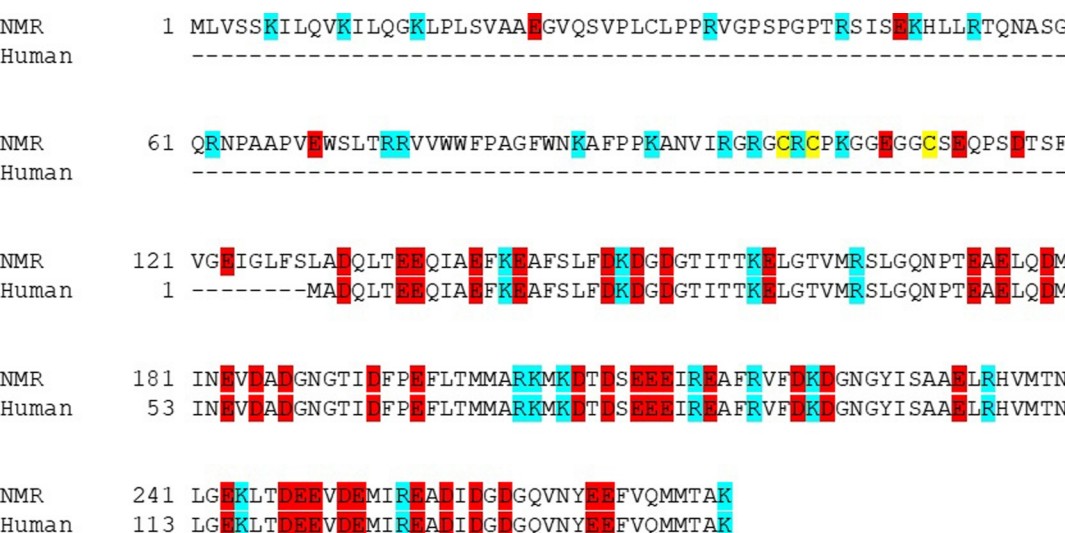

**Fig 1. Sequence alignment of naked mole rat and human calmodulins.** NMR, naked mole rat. Acidic residues are highlighted in red, basic in blue. Cysteines are highlighted in yellow.

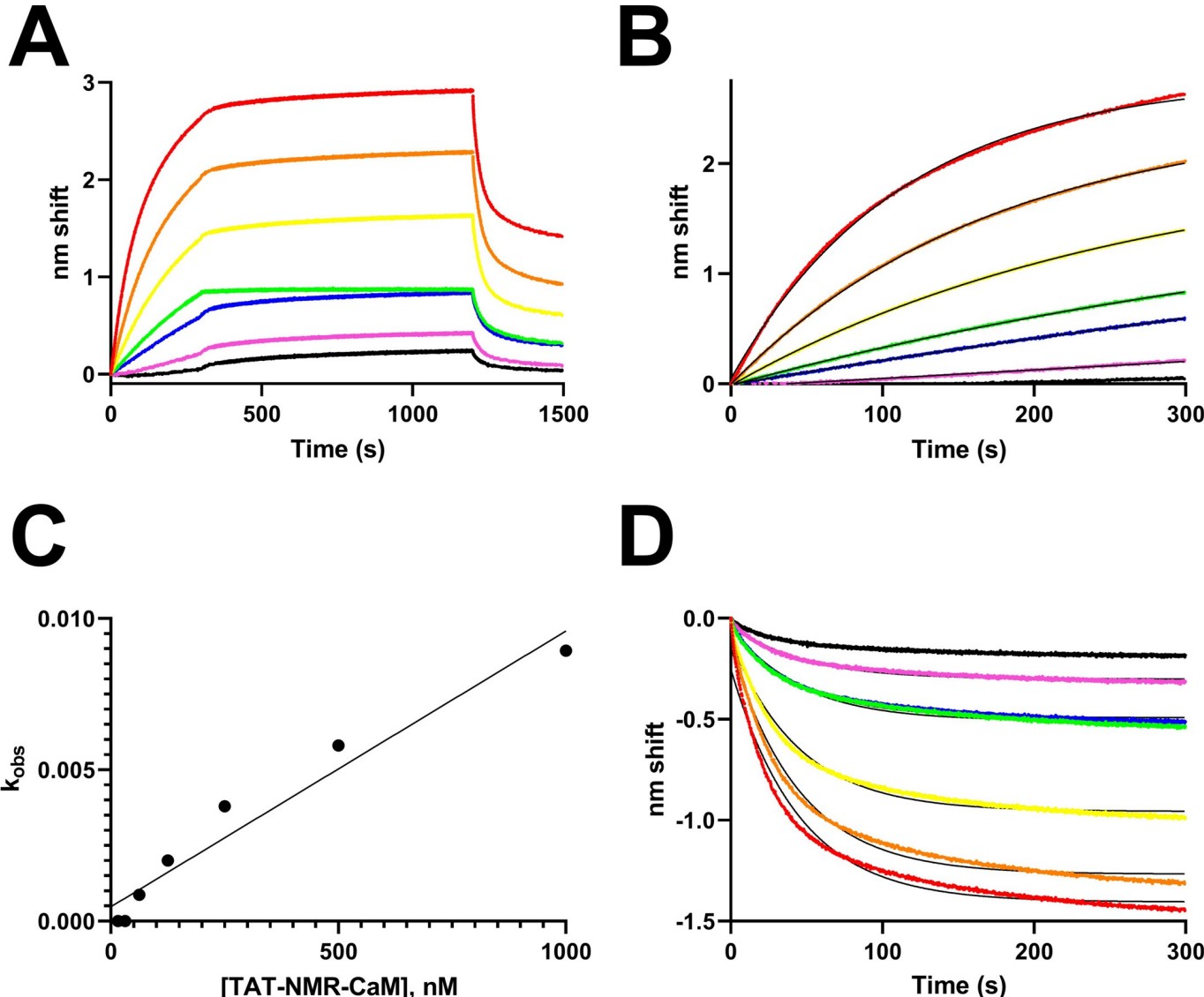

**Fig 2. BLI analysis of TAT-NMR-CaM/CBS-MBP binding.** A, sensorgram of association (0-300s), dissociation (300-1200s) and dissociation in 10 mM EDTA (1200-1500s) of sensor-tethered CBS-MBP binding to analyte TAT-NMR-CaM in which the association phase concentration was 1000 nM (red), 500 nM (orange), 250 nM (yellow), 125 nM (green), 63 nM (blue), 31 nM (magenta) and 16 nM (black). Data shown are reference subtracted against a sensor loaded with CBS-MBP run in parallel against buffer-only wells, including EDTA. B, Association phase from A fit to a single-state binding model. Raw data are colored points and fits are shown as black lines. C, $k_{obs}$ extracted from fits to B plotted against analyte concentration from which $k_a$ and $k_d$ were determined by slope and y-intercept, respectively. D, EDTA dissociation normalized to the start of the phase fit to a global single-state model.

very minor contribution of non-specific binding was ignored in kinetic calculations. Fortunately, the association phase fit reasonably well to one-state exponential models (Fig 2B), allowing determination of $k_{obs}$ for each analyte concentration. A plot of $k_{obs}$ vs analyte concentration [23] (Fig 2C) was not ideally linear, but regression analysis produced an $R^2$ of 0.95. Extracting $k_{on}$ (slope) and $k_{off}$ (y-intercept) yielded constants of 9.1 x $10^3$ $M^{-1}s^{-1}$ and 4.8 x $10^{-4}$ $s^{-1}$, respectively. The resultant $K_D$ is 53 nM. When the beginning of the EDTA dissociation phase was normalized to 0 nm shift at time 0, the resultant dissociation curves fit a global one-state exponential decay model with $k_d$ 0.02 $s^{-1}$. Because of the anomalous behavior observed, the constants as determined should not be considered rigorous kinetic assignments because

they are the result of simple one-state analyses of complex events, but can be compared with those previously determined for TAT-CaM and other CPP-EF hand adaptor proteins with other cargos [10, 11]. TAT-NMR-CaM/CBS-MBP parameters fall within the range of those observed for the other adaptor and cargo pairs. Thus it can be concluded that while nonideality in the data results from complex binding behavior whether biologically relevant or an artifact of the method, it is a small fraction of the signal observed, the constants are representative of the major binding events and show that TAT-NMR-CaM binds CBS-cargos with similar kinetics to other CPP-adaptor constructs, which is unsurprising because the NMR CaM domain is effectively invariant from the human constructs previously measured, i.e. residues 1–128 make no difference in the CaM-CBS affinity.

Cell-penetration experiments demonstrated the expected outcomes of TAT-NMR-CaM effectively delivering CBS-MBP cargos and evincing more diffuse subcellular distribution that TAT-MBP, a control protein analogous to the standard CPP-cargo linkages (Fig 3A and 3B).

TAT-NMR-CaM is thus a significant improvement relative to TAT-CaM, likely because of its overall less acidic character and separation in primary sequence of the TAT moiety and the calmodulin domain. In a future study, we intend to more thoroughly examine calmodulins from other species that have even more divergent character such as a net positive charge for their utility as CPP-adaptors.

## Real-time assay development

The vast majority of cell-penetration assays in the literature are endpoint assays either with fixed samples or in live cells, in which CPP-cargos are incubated with cells prior to imaging, usually an hour or even longer, leaving open the question of the speed at which entry and endosomal escape occur. Previous efforts from our group to examine delivery at short time points were unable to reduce cargo-addition-to-imaging time to under fifteen minutes by

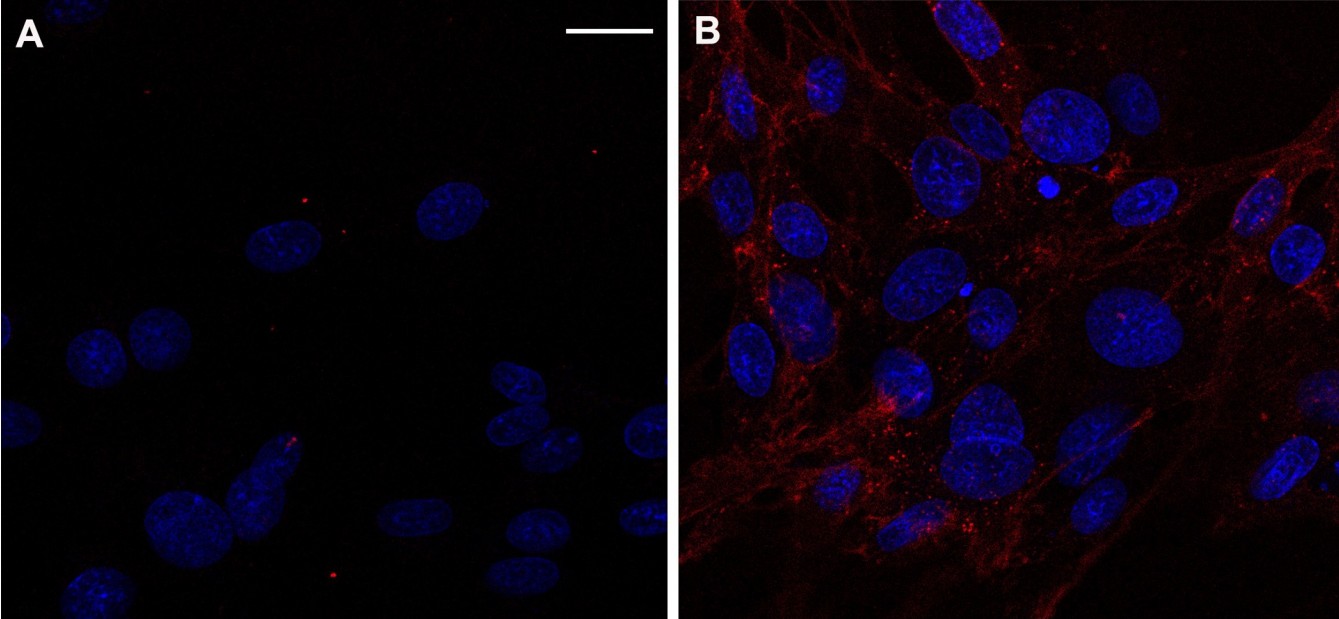

**Fig 3. TAT-NMR-CaM cell-penetration assay.** A, control with 400 nM CBS-MBP only and B, TAT-NMR-CaM/CBS-MBP (400 nM each) added to BHK cells; CBS-MBP was labelled (rendered as yellow) and cell nuclei (white) were stained prior to imaging with NucBlue. Cells were imaged with 40X objective at mid-point depth in cell and scale bar = 20 μm.

which time cargos were diffusely distributed about the cells [24]. Thus, it was very desirable to develop a real-time assay where cargo delivery could be observed *ab initio*. Because they exhibited traits that made them an improved CPP-adaptor and cargo pair, TAT-NMR-CaM and CBS-MBP were used to develop a real-time cell penetration assay. As shown in the Supporting Information, a flow system installed on an incubated confocal stage was built (S2 Fig). CPP-adaptor/CBS-cargo complexes could be delivered to a sample while imaging, allowing observation of cell penetration from time zero. BHK cells in channel slides on an incubated confocal stage were supplied with media containing adaptor cargo complexes via tubing connected to inlet and outlet points. Flow was controlled by a peristaltic pump upstream of the slide and media containing adaptors and cargos were kept in an upstream syringe reservoir until pumped.

After successful development and parameterization, the assay confirmed that TAT-NMR-CaM-based CPP-mediated delivery is rapid, with cytosolic delivery being observable in under ten minutes (Fig 4A). Real-time videos of the assays are available in Supporting Information and show noticeable delivery within seven minutes, a substantially shorter time-to-cytoplasm than prior assays allow detection. Parallel controls exhibited very little fluorescence, consistent with the labelled cargo not being delivered into cells (Fig 4B).

Real-time assays were repeated for quantitation of fluorescence. Background-subtracted fluorescence exhibited an increase over time that approximated linearity to the extent the assay was sensitive enough to determine (Fig 5). The few other studies in the literature to attempt observation at less than 30 minute time points found non-linearity, e.g. [17, 25], that is open to interpretation but may represent the attainment of equilibrium or saturation of receptors due to slow-off kinetics. Our results are not in conflict with prior findings for numerous reasons including that they are entirely different constructs. Additionally, our CPP-adaptor system is able to deliver cargos at lower concentrations, in this case 400 nM, as compared to commonly used 1–10 μM in other studies. Thus, we may be observing the CPP uptake equivalent of pseudo-first order kinetics while other studies have observed more complex phenomena related to higher CPP concentrations.

While the ultimate assay can be readily executed in a single trial, it did not arise without many iterations. Winston Churchill once said, "success consists of going from failure to failure without loss of enthusiasm." Accordingly, we conducted a great deal of enthusiastic trying and

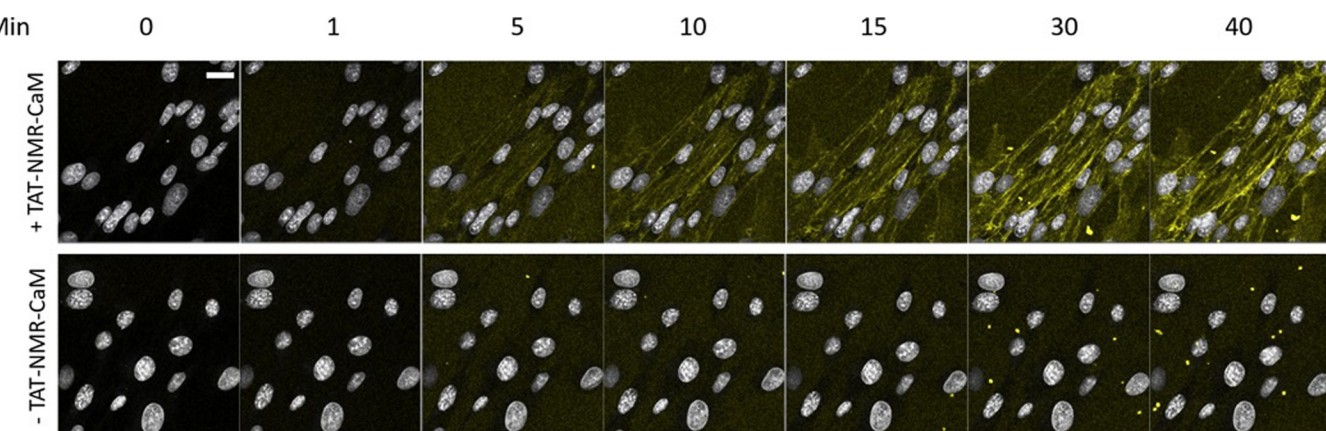

**Fig 4. Gallery of images from real-time delivery of CBS-MBP to live cells.** Top panel, TAT-NMR-CaM/CBS-MBP (400 nM each) were delivered; Bottom panel, CBS-MBP (400 nM) only was delivered. CBS-MBP was labelled (rendered as yellow) and cell nuclei (white) were stained prior to imaging with NucBlue. Cells were imaged with 40X objective at mid-point depth in cell. Scale bar = 20 μm. CBS-MBP was present in the media throughout this imaging process. Movies available in (S4–S7 Figs).

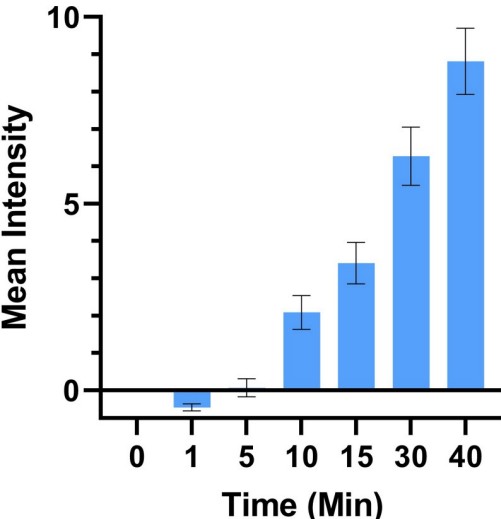

**Fig 5. Quantitation of internal cellular fluorescence as a function of time.** Background-subtracted fluorescence as a function of time. Intensities are the average of three separate experiments. Error bars represent S.E.M.

failing prior to arriving at a workable protocol. Cell stress caused by shear stress and pH variance due to the restrictive geometry was problematic. Complex delivery was originally accomplished by gravity flow from a syringe suspended above the stage, a peristaltic pump was added to control flow rate and reduce shear stress. Though done in an incubator system on the stage, concerns about lack of control of $CO_2$ concentrations in the flow system resulted in addition of HEPES to the media to buffer. These changes results in improvements of cell health as observed by morphology and the lack of compromised cells appearing to take up fluorophores in all conditions. Photobleaching from repeated imaging of cells in the same field, a common problem in real-time imaging [26, 27] was observed in preliminary experiments. Images from separate areas of the slide that had not been exposed to repeated excitation evinced greater fluorescence than the area subject to repeated imaging. Photobleaching is thus a source of error in quantitation, but has the effect of lower observable mean fluorescence and thus a resultant undermeasurement of delivered cargo.

A pulse-chase style delivery experiment was performed to follow the fate of delivered cargo. 400 nM TAT-NMR-CaM/CBS-MBP complexes were delivered to cells and left in the chamber for ten minutes after which fresh media not containing adaptor-cargo complexes were pumped into the chamber to flush out undelivered adaptor-cargos. As expected, mean fluorescence rose quickly, peaking at 15 minutes (Fig 6). Later timepoints showed no significant decline in fluorescence, indicating that delivered cargo remained within the cell. The distribution of fluorescence likewise suggests that it is largely cytoplasmic, indicating that delivered cargo was not rapidly degraded or recycled to the surface.

The TAT-CaM architecture was developed to enhance endosomal escape by post-endocytic dissociation of cargo from CPP. To quantify the difference between it and a standard CPP-cargo linkage in efficiency of cargo distribution throughout the cell, endpoint assays of MBP delivered as CBS-fused cargo via TAT-NMR-CaM or covalently conjugated to TAT ("TAT-MBP") were performed. With the hypothesis that endosomally trapped cargos would exhibit a more punctate distribution and therefore have a distribution that evinced shorter overall distances, each fluorescent pixel was measured with respect to all other fluorescent pixels within a cell. Representative micrographs for TAT-MBP and CBS-MBP delivered via

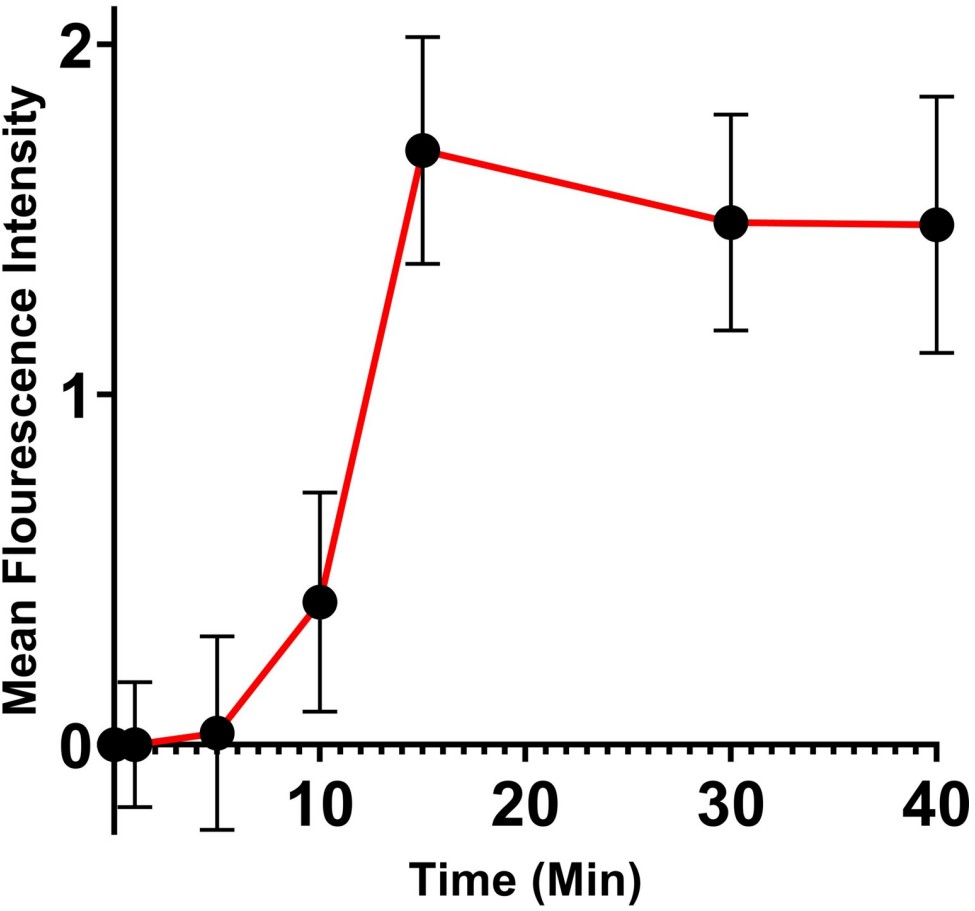

**Fig 6. Pulse-chase cell-penetration assay.** Quantification of Pulse Chase CPP Mediated Cargo Delivery in BHK21 Cells. WT TAT-NMRCaM and CBS-MBP were delivered at a concentration of 400nM for 10 minutes. Experiments completed to N = 3. Mean intensity values were averaged across all three movies. Errors bars were calculated from Standard Error of the Mean for each timepoint and were averaged across all three movies. Quantification performed using Fiji (ImageJ) software. Intracellular fluorescent signal was calculated and subtracted from extracellular background signal present at each timepoint. The red line is meant to be an aid for the eyes and does not indicate any fit to the data. Error bars represent S.E.M. for each timepoint.

TAT-NMR-CaM are shown in Fig 6A and 6B, respectively. Distributions of distances are shown in Fig 7C. As expected, the distances between pixels in cells with delivered CBS-MBP were longer than TAT-MBP, indicating broader cytosolic distribution. A Kolmogorov-Smirnov test showed significant difference (p-value < .0001). Distance analysis was not performed for other experiments in this study due to resource limitations, but we observed analogous punctate localization versus wider, presumably cytoplasmic, dispersion with TAT-myoglobin and TAT-CaM/CBS-myoglobin, a model cargo used in prior studies [10, 11]. We hope to develop an automated analysis in a future study.

To examine the potential mechanism of endocytosis, several small molecule inhibitors were applied to cells prior to delivery of TAT-NMR-CaM/CBS-MBP complexes. Chlorpromazine hydrochloride (CPZ), methyl-β-cytodextrin (MβCD) and 5-(N-ethyl-N-isopropyl) amiloride (EIPA), inhibitors of clathrin-mediated endocytosis, caveolae-mediated endocytosis (CvME) and micropinocytosis, respectively, were used in cell penetration assays at concentrations previously shown to inhibit delivery of other CPP-cargos [28–30] (Fig 8). In the presence of CPZ, protein uptake did not appear to change significantly. In some cells, small differences such as

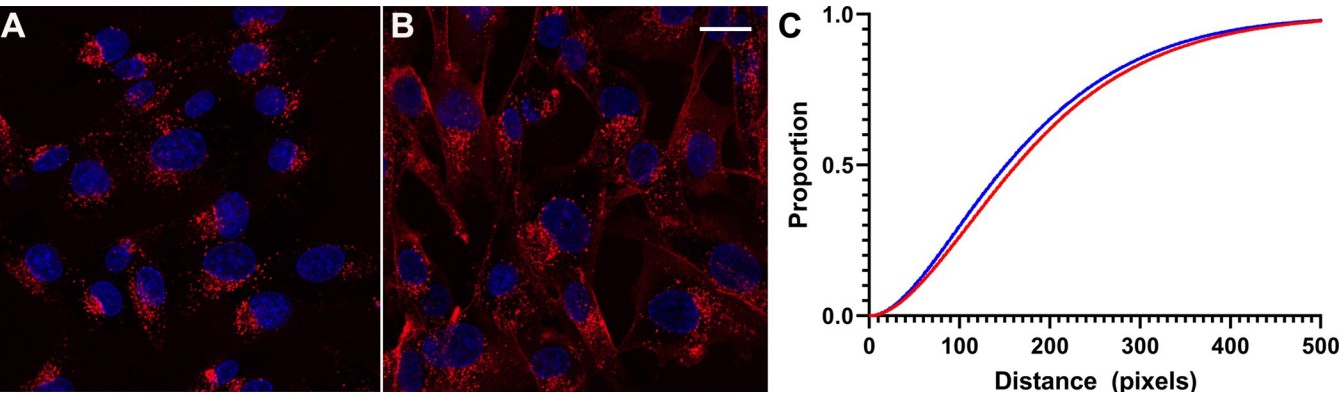

**Fig 7. Distance analysis.** A, TAT-MBP only and B, TAT-NMRCaM/CBS-MBP) complexes are representative images. C, distribution of (Blue) TAT-MBP and (Red) TAT-NMRCaM + CBS-MBP pairwise distances of fluorescent pixels are compared using the Kolmogorov-Smirnov Test for significance. Analysis completed using SAS® Studio software. Data set 10% of original size and values of "0" removed. X-axis = distance values where adjacent pixels have a distance value of 1. Y-axis = proportion of values, e.g. proportion of 0.4 = 40th percentile of data set.

the presence of larger fluorescent endosomes can be observed. Similar effects were seen in cells that were inhibited by EIPA. It is thus possible that these inhibitors were able to decrease overall protein uptake and slow endocytic trafficking during delivery, but no concrete conclusions can be made on the effectiveness of these inhibitors in reducing delivery.

Inhibition with MβCD showed a slight decrease in the number of fluorescent endosomes in confocal images. Additionally, cells appeared to have an increased level of labeled protein adhered to the membrane surface of the cell, suggesting the possibility of reduced delivery, but there are several caveats to consider when making such inferences. First, the concentration of MβCD used to inhibit CvME was very high, which could lead to other cellular changes, indirectly reducing total protein uptake. Second, due to the lower solubility of this compound, a sizable amount (10%) of $H_2O$ (solvent) was present as cells were inhibited with MβCD. It is possible that this change in environment could cause cells to have an altered uptake efficiency and total uptake level. Lastly, it is possible that the inhibitor used was unspecific and therefore was affecting other pathways in addition to CvME. With these factors in mind, no definite conclusions can be made with regard to TAT-NMR-CaM-mediated delivery, though an optimistic speculation could be that delivery is so efficient that it remains relatively unaffected at inhibitor concentrations that are known to affect other CPP-cargos.

The advantages of the real-time assay over static endpoint assays are numerous. Most importantly, it allows for real-time detection at depth in living cells even from before exposure to adaptor-cargo complexes, which also allows imaging of the same living cells before and after cargo delivery. Our prior efforts to perform fast endpoint assays achieved image collection no faster than fifteen minutes post-exposure [24], and faster non-confocal imaging is subject to artifacts due to surface adherence. Other advantages include elimination of the need for laborious and potentially stressful to the cells and artifact-inducing washing steps, performance of pulse-chase analyses, which may allow for tracking cargo and CPP-adaptor fates, including recycling to the surface or lysosomal degradation, and control of dosing with respect to replenishment of extracellular adaptor-cargo complexes.

While confident that the assay can be iteratively improved and analysis automated and made more user-friendly in time, we have chosen to publish the work in present form in the hopes that it will be of use to the field in addressing the numerous questions of kinetics, mechanism and other parameters of CPP-mediated cargo delivery. The assay is relatively facile and while requiring a confocal microscope, otherwise consists of commercially available chamber

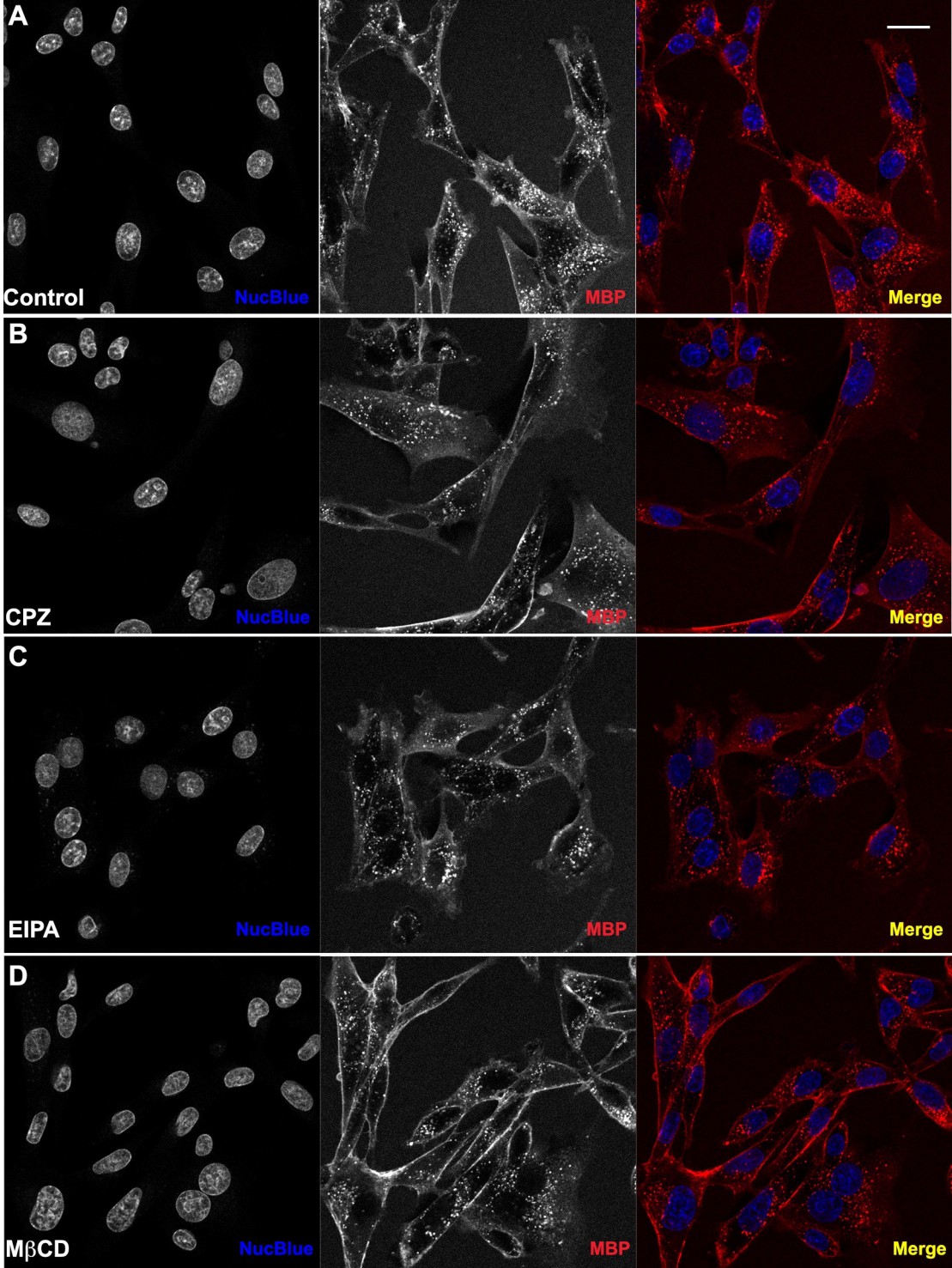

**Fig 8. Cell-penetration assays in the presence of inhibitors of endocytosis.** WT TAT-NMR-CaM (1uM) and CBS-MBP (1uM) delivered to BHK21 cells in endpoint static assay (N = 3). Cell nuclei (blue) stained with NucBlue and CBS-MBP (red) labeled with DyLight 650. Cells pretreated with inhibitor for 30 minutes. Proteins introduced to cells in presence of inhibitor. (A) No inhibitor. (B) Chlorpromazine Hydrochloride mediated inhibition of CME. (C) Methyl-β-cyclodextrin mediated inhibition of CvME. (D) 5-(N-Ethyl-N-isopropyl) amiloride mediated inhibition of MP. Cells imaged with 40X objective and scale bar = 20 μm. Red intensity values increased to 45,000 on histogram in Zen Software to improve visualization of uptake.

slides, a peristaltic pump,everyday tubing and a syringe. Fluorescence labeling was accomplished using commercially available reagents and uses chemistry generally applicable to any protein. Using the assay, it was demonstrated that our CPP-adaptor delivered a protein cargo within minutes, that delivery under the conditions examined was linear in rate and that the CaM-CBS linkage effected more diffuse cellular distribution of cargo than an analogous covalently linked cargo, consistent with more effective endosomal escape. Further, the adaptor based on calmodulin from *Heterocephalus glaber* is an improvement upon the previously described TAT-CaM and suggests that further improvements in efficiency and perhaps broader utility with respect to different cargos can be engineered by modulating the physicochemical characteristics of the CPP or CPP-adaptor, though a systematic study of cargo-specific effects with any CPP has yet to be done. Future studies will examine these issues as well as the parameters of delivery and mechanism of entry.

## Supporting information

**S1 Fig. Descriptions of plasmids and proteins encoded thereby.**
(DOCX)

**S2 Fig. Real-time imaging assay set-up on Zeiss LSM 900 confocal microscope.** A, overview of tubing system set up from outside of incubator case. B, channel slide attached to tubing system, sitting above 40X objective. C, left side view of channel slide and tubing set up. Tubing is inserted through the back of the incubator and run around the back side of the microscope. D, right side view of channel slide and tubing set up. Tubing runs from reservoir, through the back of the incubator case, and up to the peristaltic pump. Tubing runs through the pump and comes back through the back side of the incubator where it is run around the back of the microscope to the left side of channel. Right side of channel runs via tubing to waste.
(TIF)

**S3 Fig. Sensorgram demonstrating negligibility of non-specific binding in BLI.** BLI experiment in which 1 μM TAT-NMR-CaM was exposed as analyte to a sensor with biotinylated CBS-MBP (red) or a sensor without any tethered ligand (blue).
(TIF)

**S4 Fig. Movies of real-time assays.** Parts 1 and 2 of real-time assay displayed in Fig 4 with CBS-MBP delivered in complex with TAT-NMR-CaM.
(MOV)

**S5 Fig. Movies of real-time assays.** Parts 1 and 2 of real-time assay displayed in Fig 4 with CBS-MBP delivered in complex with TAT-NMR-CaM.
(MOV)

**S6 Fig. Movies of real-time assays.** Parts 1 and 2 of real-time assay control of CBS-MBP delivered without TAT-NMR-CaM.
(MOV)

**S7 Fig. Movies of real-time assays.** Parts 1 and 2 of real-time assay control of CBS-MBP delivered without TAT-NMR-CaM.
(MOV)

## Author Contributions

**Conceptualization:** Daniel P. Morris, Jonathan L. McMurry.

**Data curation:** Xuelei Ni, Stephanie A. Hill, Lydia R. Wade, Jonathan L. McMurry.

**Formal analysis:** Schuyler B. Gentry, Scott J. Nowak, Xuelei Ni, Jonathan L. McMurry.

**Funding acquisition:** Jonathan L. McMurry.

**Investigation:** Schuyler B. Gentry, Scott J. Nowak, Stephanie A. Hill, Lydia R. Wade, William R. Clark, Aidan P. Keelaghan, Daniel P. Morris, Jonathan L. McMurry.

**Methodology:** Schuyler B. Gentry, Scott J. Nowak, Xuelei Ni, Lydia R. Wade.

**Project administration:** Stephanie A. Hill, Jonathan L. McMurry.

**Supervision:** Lydia R. Wade, Daniel P. Morris, Jonathan L. McMurry.

**Writing – original draft:** Schuyler B. Gentry, Jonathan L. McMurry.

**Writing – review & editing:** Schuyler B. Gentry, Scott J. Nowak, Xuelei Ni, Stephanie A. Hill, Lydia R. Wade, Jonathan L. McMurry.

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
