## [Decision Letter · Decision Letter 0]

9 Aug 2021

PONE-D-21-20501

A real-time assay for cell-penetrating peptide-mediated delivery of molecular cargos

PLOS ONE

Dear Dr. McMurry,

Thank you for submitting your manuscript to PLOS ONE. After careful consideration, we feel that it has merit but does not fully meet PLOS ONE’s publication criteria as it currently stands. Therefore, we invite you to submit a revised version of the manuscript that addresses the points raised during the review process.

ACADEMIC EDITOR:  Dear authors,

Would you like to answer the reviewer's criticism and try to improve your manuscript according to his criticism,

We look forward to receiving your revised manuscript.

Kind regards,

Eugene A. Permyakov, Ph.D., Dr.Sci.

Academic Editor

PLOS ONE

2. Thank you for stating the following in the Competing Interests/Financial Disclosure * (delete as necessary) section:

“I have read the journal's policy and the authors of this manuscript have the following competing interests: JLM and SJN have equity interest in New Echota Biotechnology, which has exclusive license to patent US10654894B2 on the CPP-adaptor technology and applications thereof. The other authors have no competing interests. This does not alter our adherence to PLOS ONE policies on sharing data and materials.”

We note that you received funding from a commercial source: “JLM and SJN”

3. Please include captions for your Supporting Information files table at the end of your manuscript, and update any in-text citations to match accordingly. Please see our Supporting Information guidelines for more information: http://journals.plos.org/plosone/s/supporting-information.

Reviewers' comments:

Reviewer's Responses to Questions

**Comments to the Author**

1. Is the manuscript technically sound, and do the data support the conclusions?

Reviewer #1: Yes

Reviewer #2: Yes

2. Has the statistical analysis been performed appropriately and rigorously? 

Reviewer #1: Yes

Reviewer #2: Yes

3. Have the authors made all data underlying the findings in their manuscript fully available?

Reviewer #1: Yes

Reviewer #2: Yes

4. Is the manuscript presented in an intelligible fashion and written in standard English?

Reviewer #1: Yes

Reviewer #2: Yes

5. Review Comments to the Author

Reviewer #1: Major Comments: In the manuscript entitled “A real-time assay for cell-penetrating peptide-mediated delivery of molecular cargos”, the authors described a real-time method by integrating a cell containing flow chamber with a confocal microscope to allow for observation the cellular uptake kinetics of CPP-protein complex ab initio. By using this apparatus, it was found the CBS-MBP could be reversibly bonded with TAT-NMR-CaM and delivered into cells within seven minutes. The authors also performed a “distance analysis” which showed that CPP-adaptor-delivered cargo generated wider distribution throughout cells as compared to an analogous covalently-bound CPP-cargo, thus indicating the CPP-adaptor system could overcome the endosome entrapment and deliver the CBS-MBP into the cytosol. Though the apparatus-based assay showed advantages in providing a real-time analysis of the kinetic cellular uptake process, the superiority of the new method in comparison with the conventional static endpoint assay, however is less sufficiently presented. To this account, I’m afraid the current form of this manuscript is not sufficient for publication by the Plos one. And below are some major suggestions for the authors:

# 1. CPPs are valuable tools capable of transporting various pharmaceutical molecules across cell membranes, but exact penetration mechanisms of CPPs and their conjugated cargos are still not clear. We also know little about the subsequent intercellular processes of the encapsulated CPP-cargo complex, such as endosome formation, escape, and redistribution to other subcellular organelles, which are critical for bioactivity of the therapeutic molecules. This work is noteworthy to provide a real-time analysis of the kinetic process of CPP mediated delivery which is important for us to study the transporting process of CPPs. However, as a new way of observing the cellular uptake process, the “real-time assay” should be compared with the conventional static endpoint assay in a number of ways, focusing on the key factors involved in the transportation and endosomal process such as cytoskeleton, tubulin, pH change, which could help us get a better understanding of this analytic tool.

# 2. It is interesting that CPP-adaptor-delivered CBS-MBP once released from CPP-adaptor in the endosome, then the cargo could get out of the endosome and distribute into the cytosol by itself. To me it seems uncommon for a protein capable of endosome escape, so can other type of protein be excreted from endosome in the same way?

#3. There are a number of mistakes with related to figures.

1) In the manuscript, line 274-275, it is saying “CaM with no CPP in complex with CBS-MBP moiety evinced even lower background that CBS-MBP alone (Fig 3C).” However no 3C image was found in Fig.3C.

2) Line 291: “a flow system installed on an incubated confocal stage was built (S1 Fig).” it refers to Fig S2, not S1.

3) Scale bar is missing in Fig.4

Reviewer #2: The manuscript entitled “A real-time assay for cell-penetrating peptide- mediated delivery of molecular cargos” by Gentry et al report the development of assay to monitor kinetics and mechanism of the delivery of cargos to the cytoplasm with the use on calmodulin in there studies. The CPP medicate delivery was found to be linear and fast (about 7 minutes). A major bottle neck of endosomal entrapment of CPP-cargos has been discussed and troubleshoted using various analytical data. Manuscript is well written with appropriate citations.

6. PLOS authors have the option to publish the peer review history of their article (what does this mean?). If published, this will include your full peer review and any attached files.

Reviewer #1: No

Reviewer #2: No

---

## [Author Response · Author response to Decision Letter 0]

12 Aug 2021

August 10, 2021

Dr. Eugene A. Permyakov

Academic Editor

PLOS ONE

RE: Manuscript PONE-D-21-20501

Dear Dr. Permyakov,

Thank you for considering our manuscript, PONE-D-21-20501, for publication in PLOS ONE. We are appreciative of your and the reviewers’ efforts and honored that they found merit in our study and made constructive criticisms of the manuscript. We have made the suggested revisions, described in full below, and are proud to resubmit an improved, revised manuscript for further consideration. 

Our answers to editorial and reviewer points:

Editor:

1) We have done our best to ensure that our manuscript meets the style requirements.

2) Competing Interests/Financial Disclosure. We are afraid there was a miscommunication in the disclosure. While two authors do have equity interest in New Echota Biotechnology, a start-up biotechnology company that has license to a patent on the CPP-adaptor technology, the study was not funded by any commercial source at all. Thus your statement that “We note you received funding from a commercial source: JLM and SJN” is in error. JLM and SJN are the initials of the authors, Jonathan L. McMurry and Scott J. Nowak, who have equity in the company. Both are employed by Kennesaw State University and were not supported or paid by the company in any way. The company has no funding and is mostly a dormant entity at the moment. We have amended the Competing Interests statement to eliminate the use of initials to avoid the miscommunication:

I have read the journal's policy and the authors of this manuscript have the following competing interests: Authors McMurry and Nowak have equity interest in New Echota Biotechnology, a start-up company which has exclusive license to patent US10654894B2 on the CPP-adaptor technology and applications thereof. The other authors have no competing interests. This does not alter our adherence to PLOS ONE policies on sharing data and materials.

Please let us know if you need additional changes. 

3) We have added captions for the Supporting Information files as requested.

4) We have eliminated the mention of data not shown, in one case providing a new figure in Supplemental Information and in the other referring to the paper with the data that was not shown.

Reviewer #1:

1) Reviewer #1 notes in “Major Comments” that the superiority of the real-time assay to the static endpoint one is insufficiently presented. We have added a summary paragraph that more clearly describes the advantages of our method, lines 380-396.

2) The reviewer points out that the penetration mechanisms remain unclear (indeed they are a matter of great debate in the field) and implies that we should repeat the experiments “focusing on the key factors,” which include ”cytoskeleton, tubulin, pH change” and others. We completely agree that these and other factors are necessary to understanding the mechanisms and kinetics of CPP-mediated entry and delivery of cargo, but ask why is it we, of all the thousands of CPP studies are to be held to knowing the mechanisms that have for decades eluded understanding? We would instead plead that the study warrants publication now as it stand precisely so that other researchers with the resources to conduct those studies might incorporate our or similar assays in their work to better understand exactly these factors. We note that we did focus on some of the factors in our small molecule inhibitors experiments shown in Fig. 8 and the negative results we obtained strongly suggest that the mechanistic answers are likely very complex and will yet be a long time coming. We will continue to pursue them, but the scope of what the reviewer implies need be done is well beyond our ability to do at present and will require the efforts of the entire field. Best that they know about our assay now, we think. We hope the reviewer agrees. 

3) Reviewer asks if other types of proteins can be excreted from the endosome. The brief answer is yes. Endosomal escape is recognized as the largest technical hurdle to be overcome in the development of CPP-based therapeutics. We direct him or her to our prior papers describing our system (references # 10 and 11, and our review on the subject, #6). Salerno et al. demonstrates delivery of three cargo proteins that vary in size and oligomerization. In Ngwa, et al., we delivered a model cargo protein to the nucleus, endoplasmic reticulum and lysosomes using appropriate signal peptides. The reviewer is absolutely correct that it is unclear how proteins escape the endosome, but we have shown that they do (i.e. how else would they wind up in the nucleus?). The distance analysis presented in the present study represents an advance in that we can now, at any moment in time during the course of a real-time assay, quantify how much better our CPP-adaptors deliver cargo than the standard linkage; it can be a tool to measure endosomal escape going forward. We have modified comments in the last paragraph to address this critique.

4) Figure mistakes: we have corrected them according to the reviewer’s comments. We also apologize as these are the sorts of errors we endeavor never to make, but somehow usually do. Thanks to the reviewer for catching them and sorry to have made doing so necessary.

Reviewer #2: Reviewer #2 had no concerns that needed addressing in this letter.

We also made a very few other minor edits where we caught typos or realized we could enhance clarity. We also neglected to put an umlaut in Göettingen in the materials and methods sections. The changes are marked on the track changes document. 

Again, we thank you and the reviewers for your efforts. They have resulted in a substantially improved manuscript we hope you will find worthy of publication.

Sincerely,

Jonathan McMurry, Ph.D.

Corresponding Author

---

## [Editor Report · Decision Letter 1]

16 Aug 2021

A real-time assay for cell-penetrating peptide-mediated delivery of molecular cargos

PONE-D-21-20501R1

Dear Dr. McMurry,

We’re pleased to inform you that your manuscript has been judged scientifically suitable for publication and will be formally accepted for publication once it meets all outstanding technical requirements.

Kind regards,

Eugene A. Permyakov, Ph.D., Dr.Sci.

Academic Editor

PLOS ONE
---

## [Editor Report · Acceptance letter]

19 Aug 2021

PONE-D-21-20501R1 

A real-time assay for cell-penetrating peptide-mediated delivery of molecular cargos 

Dear Dr. McMurry:

I'm pleased to inform you that your manuscript has been deemed suitable for publication in PLOS ONE. Congratulations! Your manuscript is now with our production department. 

Kind regards, 

on behalf of

Prof. Eugene A. Permyakov 

Academic Editor

PLOS ONE